# Anxiolytic-like Effect Characterization of Essential Oil from Local Lavender Cultivation

**DOI:** 10.3390/ph18050624

**Published:** 2025-04-25

**Authors:** Sol Micaela Angulo, Victoria Belén Occhieppo, Cristian Moya, Rosana Crespo, Claudia Bregonzio

**Affiliations:** 1Instituto de Farmacología Experimental Córdoba (IFEC-CONICET), Departamento de Farmacología Otto Orsingher, Facultad de Ciencias Químicas, Universidad Nacional de Córdoba, Córdoba X5000, Argentina; victoria.occhieppo@unc.edu.ar (V.B.O.); rosana.crespo@unc.edu.ar (R.C.); 2Asociación Civil Foro de los Ríos, Villa General Belgrano, Córdoba X5194, Argentina; cristian.moya@unc.edu.ar; 3Grupo Cambio Rural INTA “Lavandas y Aromáticas del Valle de Calamuchita”, Villa General Belgrano, Córdoba X5194, Argentina

**Keywords:** anxiety, lavender essential oil, behavior, *Lavandula burnatii*

## Abstract

**Background:** Anxiety disorders have a 7.3% worldwide prevalence and, considering the long period of treatment, developing new efficient and safer pharmacological tools is critical. Essential oils consist of highly concentrated lipophilic compounds from plants with therapeutic potential effects, such as *Lavandula burnatii*, produced in Córdoba, Argentine, with high levels of active pharmaceutical ingredients in its essential oil (linalyl acetate and linalool). The evidence indicates that lavender essential oil could induce anxiolytic effects; however, more systematic studies are needed. **Methods:** To test the anxiolytic attributes of *Lavandula burnatii*, male Wistar rats (200–260 g) were injected intraperitoneally with two different doses of essential oil (30/80 mg/kg) or vehicle (Myritol 318, a high-purity vegetable oil), once (acute) or for 7 days. One hour after the last administration, the anxiolytic effects were evaluated using the following behavioral tests: the dark–light test and the elevated plus maze test. The open-field test was used to assess locomotor activity. **Results:** Our results showed that the lower dose of lavender essential oil induces anxiolytic effects since it increases the time spent in the aversive compartment in each evaluation. The acute administration has no impact on the behaviors evaluated. The higher dose is comparable with the control group and does not show significant differences. **Conclusions:** More studies are needed to better characterize the beneficial effects of this essential oil for anxiety disorders and to establish an adequate dosage range.

## 1. Introduction

The global prevalence of anxiety disorders, based on data from 87 studies conducted in 44 countries, has been estimated at 7.3% [1,2,3]. Furthermore, they are associated with significant disability [4,5], an increased risk of developing chronic diseases [6], and high rates of comorbidity with other neuropsychiatric disorders, especially those related to mood and substance abuse [7]. These disorders have a significant economic impact on society, primarily affecting the working-age population (18–64 years) and those with other psychiatric conditions [8,9,10].

Conventional treatment for anxiety includes benzodiazepines and selective serotonin reuptake inhibitors; however, these drugs can cause adverse effects, increased tolerance, and withdrawal symptoms upon discontinuation [11]. Since anxiety disorders are often chronic and require long-term treatment, it is essential that the medications used meet high standards of safety and adherence. In this context, herbal medicine emerges as a natural alternative with fewer side effects that can be used as a complementary or supportive treatment for anxiety disorders and other mental health conditions [12]. Essential oils (EOs) have a long history in pharmaceutical sciences as natural products with diverse biological applications in the medicinal, cosmetic, agrochemical, and nutritional fields [13]. Their use in aromatherapy and phytotherapy is widely recognized, especially for their effects on reducing stress and anxiety [14] and in treating central nervous system disorders [15]. Despite their popularity, the available scientific evidence presents methodological limitations in clinical studies on EOs [16,17,18]. Therefore, further studies are needed in order to provide greater rigor and validity for the use of EOs.

Remarkably, there is scientific evidence supporting the anxiolytic and antidepressant effects of certain EOs, with lavender EO (LEO) being one of the most studied EOs for its relaxing properties and its traditional use in the treatment of anxiety, stress, and depression [19]. Organizations such as the World Health Organization, the European Scientific Cooperative on Phytotherapy, and the European Medicines Agency [20] have approved the therapeutic use of lavender for these disorders. Several studies have analyzed its analgesic [21], anti-inflammatory [22], anxiolytic [23], antidepressant [24], and sleep-enhancing effects [23]. Several authors report beneficial effects of LEO in humans; for example, pregnant women who used LEO during the second and third trimester of pregnancy reported a significant improvement in anxiety and sleep quality [25]. Moreover, an improvement in social anxiety was reported in first-year university students after LEO inhalation [26], and its beneficial effect was even seen in people diagnosed with anxiety disorders when ingesting LEO [27]. However, despite numerous publications, there are very few controlled preclinical studies with a systematic scientific methodology that demonstrate clinical efficacy and provide credibility for the effect of LEO [28]. It is important to highlight that most of the data in the literature refer to human studies and aromatherapy.

*Lavandula burnatii* (*L. burnatii*, clone Super), a hybrid resulting from the cross between *Lavandula angustifolia* (*L. angustifolia*) and *Lavandula latifolia*, is cultivated in the province of Córdoba, Argentina. This variety has a high concentration of linalool and linalyl acetate, compounds with described anxiolytic activity [29]. *L. burnatii* is characterized by its robust growth, low to moderate water requirements, and excellent adaptability to the region’s soils. Additionally, it has an advantage since it has a 20 times higher productive yield of oil per plant than *L. angustifolia* but has the same active components (linalool and linalyl acetate), alongside other components, such as limonene, perillyl alcohol, coumarin, and camphor, that are present in other species with a relative variation in their concentrations [30] in a similar proportion. There are different chemotypes of lavender plants; in particular, in Córdoba (Argentina), *L. burnatii* and *L. angustifolia* were characterized, and the quantities of their components were similar and did not vary much. But their benefit is mainly in relation to the yield per plant of EO, and in addition, the characterization of the anxiolytic effect of this variety of lavender provides novel information for scientific research since most of these studies are based on *L. angustifolia*. Based on the described characteristics, this study aims to evaluate the effect of *L. burnatii* EO on anxiety behavior in male rats, given that there are no data on this variety cultivated in our region, to contribute solid evidence on the effect of this EO in preclinical models.

## 2. Materials and Methods

### 2.1. Essential Oil

LEO extracted from the flower tops of *Lavandula burnatii* was provided by Dr. Cristian Moya (AROMAHERBA, Calmayo, Córdoba, Argentina). The plantation is located at 850 m above sea level, and the geographic coordinates are −32.031034 and −64.459524. The aerial parts of *L. burnatii* were collected during the flowering stage from Córdoba, dried, and subjected to hydrodistillation. The oil was stored in sealed vials at 4 °C.

Chemical characterization of LEO: Gas chromatography with a flame ionization detector (GC-FID) was used to qualitatively characterize the LEO used in the present study. Quantitative analyses of EOs were performed using a PerkinElmer Clarus 500 (PerkinELmer Inc., Shelton, CT, USA) equipped with a flame ionization detector and a DB-5 capillary column (30 m × 0.25 mm inner diameter and 0.25 μm film thickness). A 1 μL aliquot of LEO (1/100 *v/v* in n-heptane) was manually injected. The oven temperature was set as follows: 60 °C for 5 min, ramped to 240 °C at 5 °C/min, and held for 10 min. Nitrogen was used as the carrier gas at a flow rate of 1 mL/min. The injector and detector temperatures were 250 °C and 280 °C, respectively. The abundance of each volatile EO compound was expressed as a relative percentage obtained by peak area normalization.

### 2.2. Animals

Adult male Wistar rats (250–300 g) were obtained from the Department of Pharmacology Otto Orsingher vivarium (Facultad de Ciencias Químicas, Universidad Nacional de Córdoba, Córdoba, Argentina) and randomly housed in groups of 3 or 4 one week before the beginning of the experimental protocol. Throughout the experiment, animals were maintained in controlled environmental conditions (20–24 °C, 12 h light/dark cycle with lights on at 7 a.m.) and had free access to food and water. Behavioral experiments were conducted between 09:00 and 14:00 h.

All procedures were conducted following the NIH Guide for the Care and Use of Laboratory Animals and approved by the Animal Care and Use Committee CICUAL, Universidad Nacional de Córdoba, Argentina (EX-2023-00769898-UNC-ME#FCQ).

### 2.3. Experimental Protocol

LEO was used at two different concentrations (30 and 80 mg/kg) in a mixture of medium-chain triglycerides (Myritol 318 Pura Química, Córdoba, Argentina). The EO was stored at 4 °C, and Myritol was stored at room temperature, both away from light until use.

Evaluation of the dose: The LEO was administered intraperitoneally (i.p.) at doses of 30 or 80 mg/kg, in a volume of 300 microliters, once daily for 7 consecutive days (Figure 1). The 30 mg/kg dose was selected based on the previous literature showing its anxiolytic effect compared to Lorazepam and Diazepam [24]. The 80 mg/kg dose was chosen because the range of doses used in rats varies widely; in unpublished results from our laboratory, a 100 mg/kg dose [31] produced a sedative effect, so a lower dose was tested instead.

Evaluation of the duration of treatment: The LEO was administered in a dose of 30 mg/kg (i.p.), in a volume of 300 microliters, acutely or once daily for 7 consecutive days. Since the dose evaluation was conducted first, we chose the dose with anxiolytic effects to assess the effective protocol of administration (Figure 1).

Control rats were treated with an equal volume of the vehicle (a mixture of medium-chain triglycerides). Experiments were conducted 1 h after the single or last drug/vehicle administration (Figure 1).

### 2.4. Behavioral Test

Elevated plus maze test: The maze has two opposite arms (50 × 10 cm), crossed with two enclosed arms of the same dimensions with 40 cm high walls. The arms are connected with a central square (10 × 10 cm), giving the apparatus the shape of a plus sign. The maze was kept in a dimly lit room and elevated 50 cm above the floor. The rats were placed individually in the center of the maze, facing an open arm. Thereafter, the number of entries and time spent in the open and enclosed arms were recorded during the next 5 min. An arm entry was defined when all four paws of the rat were in the arm [32]. 

Dark–light test: The apparatus comprises a transparent section, called the “illuminated area” (27 × 27 × 30 cm), and a smaller, fully enclosed section, called the “dark area”, (18 × 27 × 30 cm) separated by a partition with a small opening (12 × 5 cm). The light side is illuminated by a bulb. The test begins with the animal placed in the dark area, and the latency to enter the illuminated area is recorded, considering only whole-body entries. The test was measured for 5 min.

Open-field test: The apparatus (61 × 61 × 50 cm) was made of plywood and consisted of squares; the entire apparatus was painted gray and divided the floor into 16 squares. The animals were placed in the center of the apparatus and the ambulation (distance traveled) was measured, for 5 min, using Smart 3.0 software (Panlab, Barcelona, Spain) [33].

### 2.5. Statistical Analysis

Data were analyzed using one-way ANOVA and Tukey’s post hoc test. Data are reported as means ± SEM. A *p*-value of *p* < 0.05 was considered significant. If significant differences were observed in the post hoc test, the Cohen analysis was performed between the dose/treatment group and the control group (confidence interval: 95%). The analyses were performed using GraphPad Prism^®^ 8.02 software (GraphPad Software, Inc., San Diego, CA, USA), the Cohen analysis was assessed using Jamovi 2.6, and the images were made by using Inkscape^®^ 0.48 software (Free and Open-source Software licensed under the General Public License. Free Software Foundation, Inc., Boston, MA, USA).

## 3. Results

### 3.1. Chemical Composition of Lavandula burnatii Essential Oil

The geographical environment significantly influences the chemical composition of EOs in plants. Factors determining the chemical composition of EOs include temperature, altitude, rainfall, and harvest season (Table 1).

### 3.2. Effect of Lavandula burnatii Essential Oil on Elevated Plus Maze Test

Regarding the dose evaluation, the one-way ANOVA reported a significant difference in the time spent in open arms, where the 30 mg/kg dose of LEO led to significantly increased time spent in open arms compared with the control group and 80 mg/kg dose (Figure 2A; Table 2A). The time spent in open arms in the 80 mg/kg of LEO group was comparable to that of the control group and did not show a significant difference. LEO did not induce significant changes in the time spent in the closed arms since neither the 30 or 80 mg/kg dose was different from the control group (Figure 2A-Inset; Table 2A).

Moreover, the 7-day protocol (30 mg/kg of LEO) led to increased time spent in open arms (Figure 2A; Table 2B), statistically different from the acute protocol and the control groups. No significant differences were observed between these last two groups. One-way ANOVA showed no significant differences in the time spent in closed arms between the groups (Figure 2A-Inset; Table 2B).

### 3.3. Effect of Lavandula burnatii Essential Oil on Dark–Light Test

One-way ANOVA showed significant differences. A dose of 30 mg/kg of LEO showed a significant increase in the time spent in the illuminated area when compared to a dose of 80 mg/kg and the control group. One-way ANOVA indicates no significant differences were observed between the dose of 80 mg/kg and the control group (Figure 2B; Table 2C). Both doses did not affect the total number of entries in the dark area, showing no statistically significant differences when compared with the control group (Figure 2B-Inset; Table 2C).

The 7-day protocol showed a statistically significant increase in the time spent in the illuminated area compared with the acute treatment and the control group. The time spent in the illuminated area in the acute treatment group was comparable with the control group since no statistical difference was observed (Figure 2B; Table 2D). Both protocols of LEO administration (acute and 7-day) did not show a significant difference compared with the control group in terms of entries into the dark area (Figure 2B-Inset; Table 2D).

### 3.4. Effect of Lavandula burnatii Essential Oil in Open-Field Test

One-way ANOVA indicates no significant differences in the locomotor activity after LEO administration independently of the dose and protocol of administration (Figure 2C; Table 2E,F).

## 4. Discussion

In the present study, we showed clearly the anxiolytic effect of *L. burnatii* EO in rats; moreover, we presented evidence of an effective dose and two protocols of administration. Our findings were obtained using two validated tests for anxiety. The plus maze test evaluates fear of open spaces, and the dark–light test evaluates aversion to illuminated spaces [32]. In this sense, we found that a dose of 30 mg/kg induced a clear anxiolytic effect in the plus maze test, observed as an increased time spent in the open arms; meanwhile, the higher dose showed no significant impact. Moreover, the lower dose was effective under the 7-day protocol, and no significant differences were observed with the acute administration. Similar results were obtained when the animals were evaluated in the dark–light test. No significant difference was observed regarding the locomotor activity in either test, supporting the hypothesis that the difference is in the decreased fear/anxiety in the conflict test. This result is also supported by the open-field evaluation, showing no differences in locomotor activity.

Over time, these behavioral models have gained validation, largely due to consistent behavioral effects observed following the administration of benzodiazepines, particularly Diazepam [34,35]. A recent meta-analysis by Rosso et al. (2022) [36] summarized the reliability of widely used behavioral tests for anxiety in mice. They concluded that the elevated plus maze and light–dark tests are generally sensitive to anxiolytic compounds and exhibit false-negative rates near the recommended upper limit of 0.2.

Behavioral tests are commonly employed to evaluate anxiety and the potential anxiolytic effect of LEO in rats [37,38,39]. These tests are grounded in the concept of conditioned fear and are designed around approach–avoidance conflict paradigms [40]. The elevated plus maze [41] and light–dark tests [42] are widely used among standard behavioral tests. Both rely on rodents’ inherent conflict between their curiosity to explore and their natural aversion to open, brightly lit spaces. Thus, the effectiveness of anxiolytic agents is assessed based on their capacity to counteract this anxiety-induced inhibition of exploration. The open-field test is another widely used paradigm in rodent studies [39]. The total distance traveled is frequently analyzed as an indicator of locomotor activity, helping to evaluate the impact of sedative or stimulant compounds [43].

The results obtained in the present study are consistent with the data reported by Kumar (2013) [23], using *L. angustifolia* EO (Silexan^®^) with the same dose and protocol of administration. Regarding the lack of an anxiolytic effect with the highest dose (80 mg/kg), we can speculate that we are observing a dose-dependent response. Moreover, the inhalation of LEO exerts bidirectional effects, such as anxiolytic effects, or even acts as a stressor, depending on the dose, the administration methods, and the basal mood state of the individual [44]. This response seems to be similar to the well-described “U-shaped effect” for several drugs, where doses that are too high or too low may not cause any effect or even be harmful [45].

The LEO chemotype of *L. burnatii* is not very different from that of *L. angustifolia*, since the values of linalool and linalyl acetate are similar, which have shown evidence of having anxiolytic effects. In the same way, cineole [46,47] has a proven anxiolytic effect, unlike camphor, which increases neuronal activity [48]. However, since they have a majority of anxiolytic components, a synergism could be generated between them, which would result in the anxiolytic effect that we are observing. The compounds that could generate anxiety or have a counterproductive effect on anxiety are present in smaller proportions.

The beneficial effects of LEO on the central nervous system have been related to two of its main components, linalyl acetate and linalool. The results of Lopez et al. [49] showed that *L. angustifolia* EO and its main therapeutic molecules (linalool and linalyl acetate) interact with glutamate receptors (NMDA), inhibiting them and generating an anxiolytic effect. Other authors found similar effects of LEO on the glutamatergic system [50,51,52,53]. Also, it has been reported that the hydroxyl group of linalool can bind to the serotonin transporter, which could explain the antidepressant effect reported in some works [54,55]. Moreover, it has been reported that Silexan^®^ presents binding potential to serotonin-1A receptors, where its activity was significantly reduced after the ingestion of LEO [56]. Schuwald et al. [57], using Silexan^®^, identified that its potent anxiolytic effect is due to the inhibition of voltage-gated calcium channels in synaptosomes, observed through primary hippocampal neurons and cell lines with stable overexpression. The same finding was confirmed with linalool in snail neurons, in addition to increasing potassium currents [58]. Other studies suggest that LEO reversibly inhibits GABA-induced signals. Milanos et al. [59] showed that only oxygenated metabolites of linalool at carbon 8 positively influenced GABAergic currents, and hydroxylated or carboxylated derivatives at carbon 8 were ineffective. Meanwhile, acetylated derivatives of linalool did not produce significant changes, indicating that linalool metabolism reduces its allosteric potential at GABAA receptors compared to the original linalool molecule.

Finally, it is important to note that, according to a systematic analysis by Sattayakhom et al. [18] based on data from the MEDLINE, Scopus, and Google Scholar databases, 81.43% of EO studies have been conducted in human subjects, 18.57% in animal models, and only 1.43% in vitro cell cultures. Moreover, several of the scientific studies performed on both animals and humans showed significant methodological issues, such as small sample sizes, inconsistent administration methods, and a lack of placebo or control groups. In this sense, the data reported in the bibliography concerning subjects under study, the range of doses, protocols, and route of administration are wide-ranging and controversial [31,60]. We contribute to the literature by adding new systematic evidence regarding the effects of EO on the central nervous system; however, more standardized experiments are needed to confirm the benefits of lavender for brain disorders and to better understand its pharmacological and therapeutic potential.

## 5. Conclusions

Our findings added new evidence supporting the previously described anxiolytic effects of LEO, highlighting the properties of *L. burnatii.* Despite the pharmaceutical use of LEO in Europe, as far as we know, no evidence has been reported in Latin America, and the LEO formulation has not been approved in all countries. For this reason, it becomes relevant to characterize a locally produced species of lavender, thinking of a future formulation for regional use. In this sense, our study shows, for the first time, a therapeutic approach to using *L. burnatii*, a Latin American endemic plant, proposing an adequate dosage and administration protocol for future preclinical studies to contribute to systematic studies concerning lavender’s anxiolytic effects. Additionally, we evaluated the impact of an acute dose, which had not been assessed behaviorally until now. From these results, we can infer that repeated administration is necessary to observe a behavioral effect. Further studies are needed to determine the therapeutic dosage range and different administration protocols. Additionally, more systematic research is critical to describe the mechanism of action and possible side effects of LEO.

## Figures and Tables

**Figure 1 pharmaceuticals-18-00624-f001:**
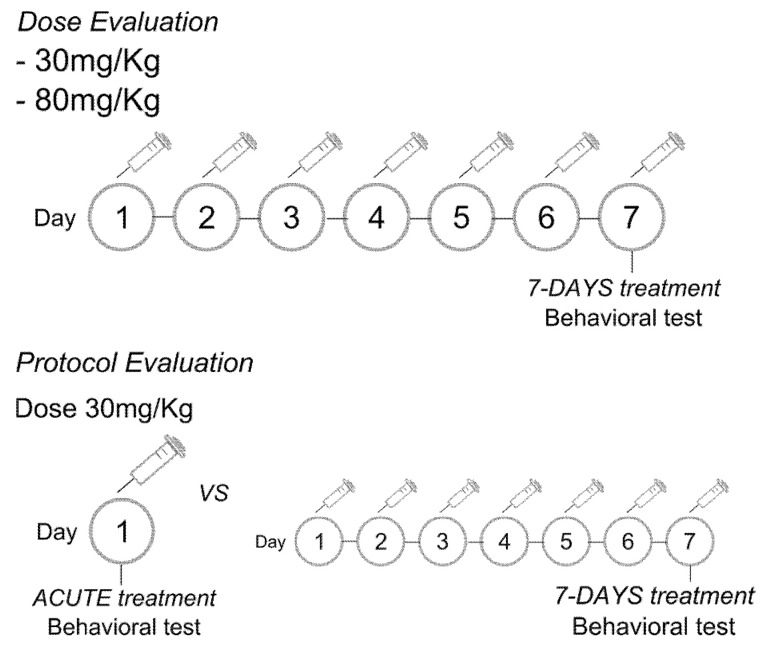
Experimental protocols used in the present study. To evaluate the dose, we tested the 30 and 80 mg/kg doses in a 7-day protocol of administration (upper scheme). For the protocol duration assessment, a dose of 30 mg/kg was evaluated after a single administration (acute) vs. the 7-day protocol of administration (lower scheme).

**Figure 2 pharmaceuticals-18-00624-f002:**
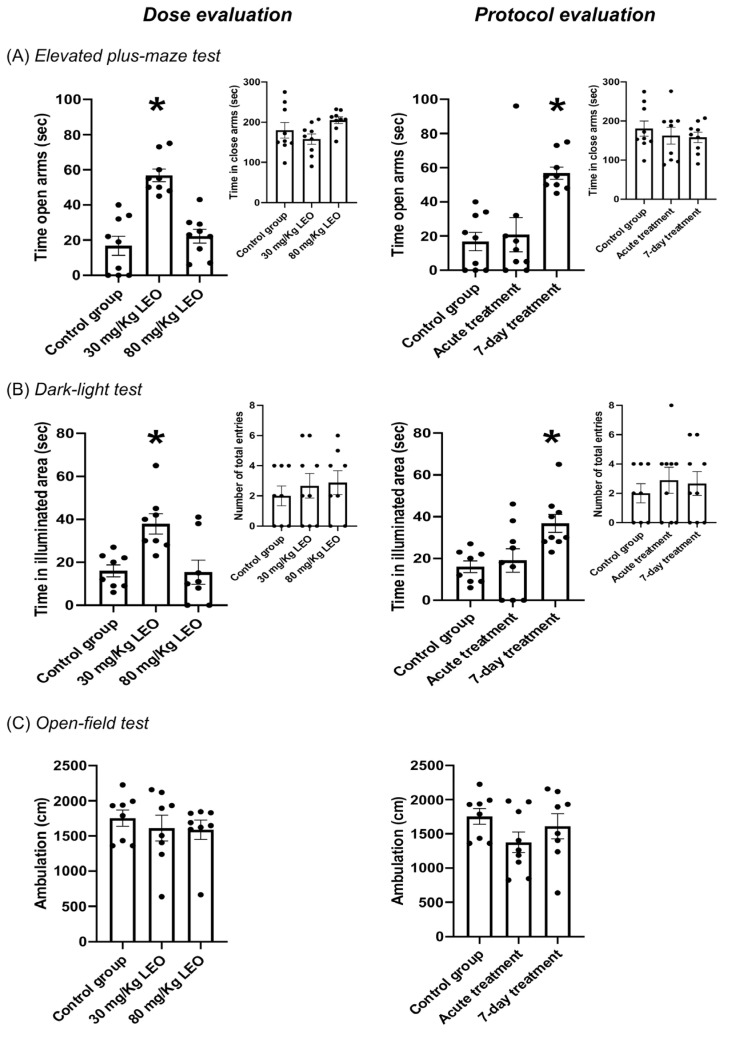
Anxiolytic effects of *Lavandula burnatii* essential oil. (**A**) Elevated plus maze test: LEO, in a dose of 30 mg/kg, induces an increase in the time (seconds) spent in open arms after a 7-day treatment regime, compared with 80 mg/kg ((**A**)-left) or the acute protocol ((**A**)-right). No significant differences were observed in the time (seconds) spent in closed arms between the groups ((**A**)-insets). (**B**) Dark–light test: a dose of 30 mg/kg of LEO leads to increased time (seconds) spent in the illuminated area after a 7-day treatment, compared with 80 mg/kg ((**B**)-left) or the acute protocol ((**B**)-right). No significant differences were observed in the total entries between the groups ((**B**)-insets). (**C**) Open-field test: no significant differences were observed between the groups in the ambulation (cm). N = 8–10 in each group. Values are mean ± SEM (* *p* < 0.05, indicates different from the all groups).

**Table 1 pharmaceuticals-18-00624-t001:** Chemical characterization of *Lavandula burnatii* essential oil through gas chromatography with a flame ionization detector.

LEO Compound	Percentage of the Sample (%)	Total in 100 µL (µL)
Linalool	25.18	25.18
Linalyl acetate	26.15	26.15
Camphor	10.03	10.03
Cineole	8.45	8.45
Borneol	3.81	3.81
Lavandulyl acetate	3.42	3.42
Caryophyllene	2.60	2.60
Limonene	1.63	1.63
β-Farnecene	1.35	1.35
β-Ocimene	1.26	1.26
4-Terpineol	0.85	0.85
α-Terpineol	0.42	0.42
Eugenol	0.20	0.20

**Table 2 pharmaceuticals-18-00624-t002:** The statistical results obtained from the behavioral tests assessed in the present work, using one-way ANOVA and Tukey’s multiple comparisons. (**A**,**B**) Elevated plus maze test for dose and protocol duration evaluation, respectively. (**C**,**D**) Dark–light test for dose and protocol duration assessment, respectively. (**E**,**F**) Open-field test for dose and protocol duration evaluation, respectively. *p* < 0.05 was considered significant.

(**A**) Elevated plus maze test—Dose		
Time in open arms		
One-Way ANOVA	F = 24.67	*p* < 0.0001
Tukey’s multiple comparisons test	*p*	
“Control group” vs. “30 mg/kg of LEO”	<0.0001	
“Control group” vs. “80 mg/kg of LEO”	0.6569	
“30 mg/kg of LEO” vs. “80 mg/kg of LEO”	<0.0001	
Cohen’s analysis	d	
“Control group” vs. “30 mg/kg of LEO”	2.91	
	Inferior = 1.53	Superior = 4.25
Time in closed arms		
One-Way ANOVA	F = 5.249	*p* = 0.0129
Tukey’s multiple comparisons test	*p*	
“Control group” vs. “30 mg/kg of LEO”	5.868	
“Control group” vs. “80 mg/kg of LEO”	0.0966	
“30 mg/kg of LEO” vs. “80 mg/kg of LEO”	0.0112	
(**B**) Elevated plus maze test—Protocol duration		
Time in open arms		
One-Way ANOVA	F = 10.21	*p* = 0.0006
Tukey’s multiple comparisons test	*p*	
“Control group” vs. “Acute”	9.117	
“Control group” vs. “7-day”	0.0011	
“Acute” vs. “7-day”	0.0032	
Cohen’s analysis	d	
“Control group” vs. “7-day”	2.91	
	Inferior = 1.53	Superior = 4.25
Time in closed arms		
One-Way ANOVA	F = 0.42	*p* = 0.6618
(**C**) Dark–light test—Dose		
Time in the illuminated area		
One-Way ANOVA	F = 8.04	*p* = 0.0026
Tukey’s multiple comparisons test	*p*	
“Control group” vs. “30 mg/kg of LEO”	0.0069	
“Control group” vs. “80 mg/kg of LEO”	0.9947	
“30 mg/kg of LEO” vs. “80 mg/kg of LEO”	0.0055	
Cohen’s analysis	d	
“Control group” vs. “30 mg/kg of LEO”	1.41	
	Inferior = 0.316	Superior = 2.47
Total entries		
One-Way ANOVA	F = 0.3463	*p* = 0.7111
(**D**) Dark–light test—Protocol duration		
Time in the illuminated area		
One-Way ANOVA	F = 6.225	*p* = 0.0069
Tukey’s multiple comparisons test	*p*	
“Control group” vs. “Acute”	0.8887	
“Control group” vs. “7-day”	0.0104	
“Acute” vs. “7-day”	0.0246	
Cohen’s analysis	d	
“Control group” vs. “30 mg/kg of LEO”	1.41	
	Inferior = 0.316	Superior = 2.47
Total entries		
One-Way ANOVA	F = 0.3188	*p* = 0.7302
(**E**) Open-field test—Dose		
Ambulation		
One-Way ANOVA	F = 0.3647	*p* = 0.6987
(**F**) Open-field test—Protocol duration		
Ambulation		
One-Way ANOVA	F = 1.603	*p* = 0.2239

## Data Availability

The original contributions presented in this study are included in the article. Further inquiries can be directed to the corresponding authors.

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
