# Peer review of "Anxiolytic-like Effect Characterization of Essential Oil from Local Lavender Cultivation"

_pharmaceuticals, 2025, doi:10.3390/ph18050624_

Round 1
Reviewer 1 Report
Comments and Suggestions for Authors
Anxiety is a widely used term to indicate a complex of cognitive, behavioral and physiological reactions that occur following the perception of a stimulus considered threatening. The pharmacological therapy of anxiety is based on anxiolytic drugs, especially benzodiazepines, widely used, effective only if used occasionally and for very short periods. Otherwise they present serious problems of addiction. Therefore, in order to reduce side effects, current research is directed towards the study of the beneficial effects of herbal medicine. In particular, in this work, the anxiolytic effect of the essential oil of Lavandula burnatii on the behavior of male Wistar rats was evaluated, which were injected intraperitoneally with 2 different doses of essential oil or vehicle (Myritol, a high-purity vegetable oil), once (acute) or for 7 days.
The study of the therapeutic effects of natural substances is a very current topic.
The Introduction section provides background information on the research topic, states the research question, and outlines the importance of the study.
The Materials and Methods section describes the study design. It specifies the data collection methods, the sample, the tools, and the statistical analysis techniques used.
The Results section presents the results of the study, using Tables and Figures to present the data resulting from the analyses.
The Discussion section interprets the results and compares them with previously published research.
Overall, the structure of the manuscript is well done, but some changes are needed, as requested below.
In particular,
Tables and Figures are fundamental elements of a scientific text and must be presented in the right way. On the other hand, Tables have the purpose of making the text faster to understand through the synthetic and organic presentation of the data; Figures must have an independent informative content, regardless of their description within the text. In this work, they must be revised.
Figure 1: The explanation of the figure must be included in the text. Furthermore, its caption must be clearer and more detailed.
Similarly, please add a short explanatory caption without the aid of text for each Table and Figure (including the various panels of Tables 2-5).
The Conclusions section should provide a brief summary of the Results and Discussion, but also be engaging, concise, and leave the reader with a clear understanding of the research. In this case, it should be revised. It is suggested to move the last paragraph of the Discussion (lines 276-279) to the Conclusions and add the limitations of the study conducted.
Insert the list of abbreviations present in the text.
Author Response
Comentarios 1 : Las tablas y figuras son elementos fundamentales de un texto científico y deben presentarse correctamente. Por otro lado, las tablas tienen como objetivo facilitar la comprensión del texto mediante la presentación sintética y orgánica de los datos; las figuras deben tener un contenido informativo independiente, independientemente de su descripción dentro del texto. En este trabajo, deben revisarse. Figura 1: La explicación de la figura debe incluirse en el texto. Además, su pie de foto debe ser más claro y detallado. De igual manera, añada un breve pie de foto explicativo sin texto para cada tabla y figura (incluidos los distintos paneles de las tablas 2 a 5).
Respuesta 1 : La respuesta está marcada en rojo en el documento. Gracias por indicarlo. Hemos añadido un título más descriptivo y mejorado la redacción para que los resultados sean más fáciles de entender.
Comentarios 2 : La sección de Conclusiones debe ofrecer un breve resumen de los Resultados y la Discusión, además de ser atractiva, concisa y permitir al lector una comprensión clara de la investigación. En este caso, debería revisarse. Se sugiere trasladar el último párrafo de la Discusión (líneas 276-279) a las Conclusiones y añadir las limitaciones del estudio realizado.
Respuesta 2 : La respuesta está marcada en verde en el documento. Estamos de acuerdo con este comentario. Muchas gracias por su comentario. Esperamos haber mejorado la discusión y haber hecho que el desarrollo de este trabajo sea más claro y atractivo.
Comentarios 3 : Insertar la lista de abreviaturas presentes en el texto.
Respuesta 3 : La respuesta está marcada en azul en el documento. Gracias por señalarlo. Muchas gracias por su comentario. En un momento dado, decidimos no añadirla porque solo hay dos abreviaturas, pero para una mejor comprensión, las añadimos como sugirió.
Reviewer 2 Report
Comments and Suggestions for Authors
Recommendations for Authors Introduction:
The introduction effectively outlines anxiety disorders and the potential benefits of essential oils. However, the research gap being addressed should be more clearly stated.
Some references are outdated; inclusion of more recent studies would strengthen the background.
Research design:
The experimental design is appropriate for evaluating the anxiolytic effects of the essential oil. However, further justification for the doses chosen (30 mg/kg and 80 mg/kg) is needed to increase the rigour of the study.
Methods:
The methods section is well described but further clarification is needed:
How were the lavender essential oil doses determined? Were they based on previous studies?
Provide more details on the selection of behavioural tests and their validation to assess anxiolytic effects in animal models.
Results:
The results are clearly presented, but the statistical analysis should be more explicit. Inclusion of effect sizes and confidence intervals where appropriate would improve clarity.
Further discussion of why the lower dose showed anxiolytic effects while the higher dose did not is recommended.
Conclusions:
The conclusions are consistent with the findings, but greater emphasis on the limitations of the study and potential future research directions would enhance its impact.
Quality of the English language:
The language is generally clear, but could be improved for conciseness and readability.
Minor grammatical errors and awkward phrasing should be revised to improve clarity and flow.
Comments on the Quality of English LanguageThe English is understandable but could be clearer and more concise. Some sentences are too long and should be restructured for better readability. Additionally, minor grammatical errors and awkward phrasing need revision.
Author Response
Comentario 1 : La introducción describe eficazmente los trastornos de ansiedad y los posibles beneficios de los aceites esenciales. Sin embargo, la brecha de investigación que se está abordando debería explicarse con mayor claridad.
Algunas referencias están desactualizadas; la inclusión de estudios más recientes fortalecería los antecedentes.
Respuesta 1 : La respuesta está marcada en azul en el documento. Estamos de acuerdo con este comentario. Muchas gracias por su comentario. Buscamos las citas más recientes posibles, pero sorprendentemente no tuvimos éxito, especialmente en lo que respeta a la prevalencia de la ansiedad. A continuación, se muestra un ejemplo de un artículo publicado recientemente, pero lo citaron del mismo año que nosotros (10.1177/09727531241266094), y otros siguen el mismo ejemplo.
Comentarios 2 : El diseño experimental es adecuado para evaluar los efectos ansiolíticos del aceite esencial. Sin embargo, se requiere mayor justificación de las dosis elegidas (30 mg/kg y 80 mg/kg) para aumentar el rigor del estudio.
Respuesta 2 : La respuesta está marcada en rojo en el documento. Gracias por señalarlo. Muchas gracias por su comentario. Hemos agregado esta información al manuscrito enviado recientemente porque la consideramos importante.
Comentarios 3 : La sección de métodos está bien descrita, pero se necesitan más aclaraciones:
¿Cómo se determinan las dosis de aceite esencial de lavanda? ¿Se basaron en estudios anteriores? Proporcione más detalles sobre la selección de pruebas de comportamiento y su validación para evaluar los efectos ansiolíticos en modelos animales.
Respuesta 3 : La respuesta está marcada en rojo en el documento. Estamos de acuerdo con este comentario. Muchas gracias por su comentario. Hemos añadido más información sobre las pruebas y la selección de dosis a la discusión.
Comentario 4 : Los resultados se presentan con claridad, pero el análisis estadístico debería ser más explícito. La inclusión de la magnitud del efecto y los intervalos de confianza, cuando corresponda, mejorará la claridad.
Se recomienda analizar más a fondo por qué la dosis más baja mostró efectos ansiolíticos mientras que la dosis más alta no.
Respuesta 4 : La respuesta está marcada con verde en el documento. Gracias por señalarlo. Muchas gracias por su comentario. Hemos añadido un análisis de Cohen a las tablas estadísticas al final del manuscrito, que proporciona información sobre la magnitud del efecto.
Comentarios 5 : Las conclusiones son consistentes con los hallazgos, pero un mayor énfasis en las limitaciones del estudio y las posibles direcciones futuras de investigación mejorarían su impacto.
Respuesta 5 : La respuesta está marcada en negro en el documento. Muchas gracias por su comentario. Hemos añadido las limitaciones en la conclusión para una mejor interpretación y calidad del trabajo.
Comentarios 6 : Calidad del inglés: El lenguaje es, en general, claro, pero podría mejorarse para mejorar la concisión y la legibilidad. Se deben revisar los errores gramaticales menores y las frases torpes para mejorar la claridad y la fluidez.
Respuesta 6 : Muchas gracias por su comentario, estamos tratando de mejorar la calidad del lenguaje en el manuscrito que envió recientemente.
Reviewer 3 Report
Comments and Suggestions for Authors
This brief report aimed to evaluate the effects of lavender (Lavandula burnatii) essential oil (cultivated in Córdoba, Argentina) on anxiety behavior in rats. This study is simple, but it brings novelty to the research field to some extent. There are, however, some concerns that need to be addressed.
The Discussion of obtained results need to be more detailed. First, the results need to be discussed in terms of the LEO chemotype used. LEO chemotypes should be mentioned in the Introduction section, as well as their specifities. That is especially important since some chemotypes are better for relaxation (linalool-rich), while others are energizing (camphor-rich) when we talk about aromatherapy for example.
Furthermore, the biphasic response of LEO and essential oils in general should be more in-depth discussed. While lavender is primarily recognized for its anxiolytic properties, certain studies highlight its potential to induce anxiety or exhibit no significant effects under specific conditions. In stress-free animals, lavender essential oil inhalation can act as a stressor, increasing anxiety-like behaviors. This bidirectional effect suggests that lavender may not always be anxiolytic and could exacerbate anxiety in low-stress environments (https://doi.org/10.1177/1934578X1200701132). Also, high concentrations of lavender oil may paradoxically increase anxiety in some individuals. It was shown that LEO exhibits a biphasic response as an anxiolytic, showing effectiveness in reducing anxiety at lower doses while potentially being less effective or even counterproductive at higher doses, depending on the context and method of application (https://doi.org/10.23947/young.2022.41-44).
Tables 2-6 are redundant; the same results can be seen in figures. These tables also do not have legends. Besides, “brief reports” are short, observational studies that usually contain two figures and/or a table (according to the journal rules).
The decimal points need to be used instead of decimal commas throughout the whole manuscript.
Eventually, the manuscript would benefit from detailed language check; there are several language-related errors that need to be corrected.
There are also some other concerns as follows:
Abstract: Specify that Córdoba is in Argentina. What do “high levels of active pharmaceutical ingredients” refer to – essential oil or plant itself? The whole sentence is not clear; please, rephrase it.
Introduction: As previously stated, the introduction of REO chemotypes is needed. The term "palliative" might not be the best choice; a more suitable phrasing could be "supportive" (line 42).
Lines 108-110: The first sentence is repeated. The second needs to be in the Materials section and not in the Experimental Protocol.
Lines 152-153: That is discussion (not results) and it is not referring to Table 1.
Table 1: add the row “total”.
Comments on the Quality of English LanguageThe manuscript would benefit from detailed language check; there are several language-related errors that need to be corrected.
Author Response
Comments 1: The Discussion of obtained results need to be more detailed. First, the results need to be discussed in terms of the LEO chemotype used. LEO chemotypes should be mentioned in the Introduction section, as well as their specifities. That is especially important since some chemotypes are better for relaxation (linalool-rich), while others are energizing (camphor-rich) when we talk about aromatherapy for example.
Response 1: Response is marked with green in the document.Thank you for pointing this out. Thank you very much for your comment. We agree with adding information about chemotypes, and we've added it to the introduction and discussion.
Comments 2: Furthermore, the biphasic response of LEO and essential oils in general should be more in-depth discussed. While lavender is primarily recognized for its anxiolytic properties, certain studies highlight its potential to induce anxiety or exhibit no significant effects under specific conditions. In stress-free animals, lavender essential oil inhalation can act as a stressor, increasing anxiety-like behaviors. This bidirectional effect suggests that lavender may not always be anxiolytic and could exacerbate anxiety in low-stress environments (https://doi.org/10.1177/1934578X1200701132). Also, high concentrations of lavender oil may paradoxically increase anxiety in some individuals. It was shown that LEO exhibits a biphasic response as an anxiolytic, showing effectiveness in reducing anxiety at lower doses while potentially being less effective or even counterproductive at higher doses, depending on the context and method of application (https://doi.org/10.23947/young.2022.41-44).
Response 2: Response is marked with red in the document. Thank you very much for your comment and the information provided; it helped us a lot to be able to prepare a correct justification.
Comments 3: Tables 2-6 are redundant; the same results can be seen in figures. These tables also do not have legends. Besides, “brief reports” are short, observational studies that usually contain two figures and/or a table (according to the journal rules). The decimal points need to be used instead of decimal commas throughout the whole manuscript.
Response 3: Thank you very much for your comment. We have improved the article with this comment on the recently uploaded manuscript.
Comments 4: Eventually, the manuscript would benefit from detailed language check; there are several language-related errors that need to be corrected.
Response 4: Thank you very much for your comment, we are trying to improve the quality of the language in the manuscript you recently submitted.
Comments 5: Abstract: Specify that Córdoba is in Argentina. What do “high levels of active pharmaceutical ingredients” refer to – essential oil or plant itself? The whole sentence is not clear; please, rephrase it.
Response 5: Response is marked with blue in the document. Thank you for pointing this out. Thank you very much for your comment. We clarify in the abstract that this refers to essential oil.
Comments 6: Introduction: As previously stated, the introduction of REO chemotypes is needed. The term "palliative" might not be the best choice; a more suitable phrasing could be "supportive" (line 42).
Response 6: Response is marked with black in the document. We agree with this comment. Thank you very much for your comment. We've added information about the chemotype to the introduction and changed it to "palliative."
Comments 7: Lines 108-110: The first sentence is repeated. The second needs to be in the Materials section and not in the Experimental Protocol.
Response 7: Thank you very much for your comment, we made the necessary change.
Comments 8: Lines 152-153: That is discussion (not results) and it is not referring to Table 1.
Response 8: Thank you for pointing this out. We made the change to improve the interpretation and clarity of the text.
Comments 9: Table 1: add the row “total”.
Response 9: Thank you for pointing this out. Thank you very much for your comment, we made the necessary change hoping that we have properly understood your suggestion.
Round 2
Reviewer 3 Report
Comments and Suggestions for Authors
Most of my suggestions have been accepted, and the manuscript has been improved.
Table 1 was changed (even values); in my opinion, the paper needs to contain the complete chemical characterization of the essential oil.
Why were p values changed? Was statistic analysis done again (with a different test)?
Comments on the Quality of English LanguageThe same as in the first revision.
Author Response
Comment 1: Table 1 was changed (even values); in my opinion, the paper needs to contain the complete chemical characterization of the essential oil.
Response 1: We have accordingly to emphasize this point. Thank you very much for your comment. We changed the values ​​because we measured two different batches of oils using GC-FID, but the values ​​are similar. We reposted the original table since we had a more detailed composition of the essential oil. We just wanted to show that the chemotype is linalool rich and not camphor rich. The values of linalool vary from 25.18 to 28.11 % but linalyl acetate goes from 26 to 42 % in L burantii and in L angustifolia in this region of Cordoba, Argentina.
Comment 2: Why were p values changed? Was statistic analysis done again (with a different test)?
Response 2: We have revised to emphasize this point. Thank you very much for your comment. We haven't made any changes to the p value. We've only added a new parameter (Cohen's D) to measure the magnitude of essential oil therapy at the request of another reviewer.
Comment 3: Comments on the Quality of English Language: The same as in the first revision.
Response 3: We have revised to emphasize this point. Thank you very much for your comment. We sent someone to review the language of the text and made the necessary changes. María José Martinez, Public Sworn Translator of English, Register Number 532, National University of Córdoba.